Genome-wide identification of calcium-dependent protein kinase (CDPK) family members in Phaseolus vulgaris L. and expression analysis during abiotic stresses

Güçlü Gülşen gulsenguclu@cumhuriyet.edu.tr
Vocational School of Health, Sivas Cumhuriyet University , Sivas , Center , Turkey
Langille Barbara
Electronic publication date: 2025 Oct 20
Publication date: 2025
Volume: 13
Electronic Location ID: e20217
Received 2025 May 26; Accepted 2025 Sep 19
Copyright: ©2025 Güçlü
Copyright year: 2025
Copyright holder: Güçlü
License: This is an open access article distributed under the terms of the Creative Commons Attribution License, which permits unrestricted use, distribution, reproduction and adaptation in any medium and for any purpose provided that it is properly attributed. For attribution, the original author(s), title, publication source (PeerJ) and either DOI or URL of the article must be cited.
License URL: https://creativecommons.org/licenses/by/4.0/

Keywords: Genome-wide, Phaseolus vulgaris L., CDPK gene, Gene expression, Abiotic stress

Funding: The author received no funding for this work.

==============================
Background

Calcium-dependent protein kinases (CDPKs) are a family of enzymes that are essential in plant signaling pathways. These kinases are activated in response to changes in calcium ion (Ca2+) concentration under stress conditions. Although CDPK gene families have been investigated in various plants, comprehensive genome-wide analyses and expression studies of CDPK genes in Phaseolus vulgaris L. under abiotic stress have not yet been performed. The objective of this research is to perform a genome-wide analysis of the CDPK gene family in common bean and evaluate the expression patterns of these genes under salt and drought stress conditions.

Methods

This study presents a comprehensive bioinformatics analysis focusing on the phylogenetic relationships, chromosomal distribution, gene structures, conserved motifs, promoter regulatory elements, and expression profiles under salt and drought stress conditions of the CDPK gene family in common bean.

Results

In this study, 25 PvCDPK genes were identified in the common bean genome. The lengths of proteins vary between 298 and 582 amino acids (aa), and their molecular weights (kDa) range from 33.43 kDa to 65.13 kDa. The majority of the PvCDPKs located on a total of eight chromosomes have six introns. Phylogenetic analysis revealed that PvCDPK proteins are grouped into three major clades along with Arabidopsis thaliana and Glycine max orthologs. The divergence times for six pairs of segmentally duplicated genes ranged from 48.94 million years ago (MYA) to 65.57 MYA, while tandem duplicates ranged from 32.09 to 84.95 MYA.

Conclusions

Comparative expression analysis of PvCDPK genes revealed varying expression levels depending on the two common bean cultivars. Furthermore, these observations suggest that PvCDPK genes could be essential for the growth and development of common beans in response to abiotic stresses such as drought and salt. This is the first study to investigate the CDPK gene family in common bean, and the identified genes obtained can be directly evaluated as candidate genes for marker-assisted selection or gene editing approaches. In addition, the findings are expected to contribute to the development of resilient cultivars capable of withstanding climate change.

Introduction

Plants are continuously exposed to biotic and abiotic stresses during their developmental process. Highly complex mechanisms are activated to respond to the effects of abiotic stresses like salinity, temperature, drought, and heavy metals (Takahashi et al., 2020; Nawaz et al., 2023). In this way, plant adaptation is enhanced, and the detrimental effects of the stress are minimized.

Secondary messengers play important roles in providing this adaptation. Calcium ion (Ca+2), one of the secondary messenger molecules, plays a critical role in regulating plant growth, development, and responses to abiotic stresses (Li et al., 2022a; Li et al., 2022b; Ray, 2017). Proteins that detect changes in the cytoplasmic Ca+2 levels initiate the phosphorylation process, thereby activating downstream signaling pathways in plants (Ravi, Foyer & Pandey, 2023; Zhao et al., 2021). Ca+2, whose concentration varies under stress, binds to the CaM-like domain to activate CDPKs, which allow the plant to react appropriately throughout growth and development (Bredow & Monaghan, 2022). Ca2+-dependent protein kinases (CDPKs), calmodulin (CaM), calcineurin B-like proteins, and CAM-like proteins (CMLs) are some of the classes of Ca2+-binding proteins in plants (Zeng et al., 2023). Particularly, CDPKs are crucial calcium-binding proteins that are only present in protists, green algae, and plants—not in fungi or animals (Hamel, Sheen & Séguin, 2014; Wang et al., 2015).

Research has shown that CDPKs are effective under stress conditions. In rice (Oryza sativa), OsCDPK4 has been reported to protect the cell membrane from oxidative damage and thus increase salt and drought tolerance (Campo et al., 2014). AtCPK28 was revealed to decode cold-affected Ca+2 signals in Arabidopsis thaliana and increase plant resistance to cold by phosphorylating Nin-Like Protein 7 (NPL7) (Shi et al., 2018; Ding et al., 2022). In another study, it was reported that the transcription of TaCDPK25-U-AS1 and TaCDPK25-U-AS2 increased under drought stress in wheat (Triticum aestivum), and this increased the drought resistance of the plant (Linghu et al., 2023). It has been revealed that FaCDPK1 and FaCDPK3, and FaCDPK4 and FaCDPK11, which are among the CDPKs in strawberry (Fragaria x ananassa), form a strong response to salt, while FaCDPK4 and FaCDPK11 form a strong response to drought, and that drought-related genes are significantly affected by ABA treatment. It was also claimed that this may affect drought-related proteins (Crizel et al., 2020).

Phaseolus vulgaris L. (common bean) belongs to the family Fabaceae, which has 640 genera, and is a member of the genus Phaseolus, which is known to have 240 species (Broughton et al., 2003; Silva et al., 2020). Common beans are a highly nutritious and economically significant plant, extensively cultivated and consumed globally. Abiotic stress factors, particularly salinity and drought, significantly contribute to crop loss in P. vulgaris L. Although CDPK genes have been studied in various plant species, comprehensive analyses of these genes in P. vulgaris L. under abiotic stresses are still limited. Therefore, the present study utilized bioinformatics data to identify and characterize the CDPK genes in common beans. Additionally, the qRT-PCR was employed to clarify the functions of these genes in response to drought and salt stress. Furthermore, this study provides molecular targets for the development of stress-tolerant common bean varieties. The characterization of these genes offers an integrative understanding of common bean metabolism. This study represents the first comprehensive investigation of the CDPK gene family in P. vulgaris L., providing valuable candidate genes for future breeding programs aimed at improving abiotic stress tolerance and climate resilience in common bean.

Materials & Methods

Identification of PvCDPK gene family and analysis of basic parameters

Amino acid (aa) sequences of the P. vulgaris L. CDPK gene family were retrieved from the Phytozome v12.1 database (https://phytozome-next.jgi.doe.gov/) under accession number PF03492 (http://pfam.xfam.org). The genomes of A. thaliana and Glycine max were examined in the same database to identify potential CDPK proteins (Lamesch et al., 2012; Valliyodan et al., 2019). The default configurations of the Hidden Markov Model (HMM) validated the CDPK protein sequences. Table S1 enumerates the CDPK protein sequences of various plants. The HMMER database (http://www.ebi.ac.uk) was utilized to examine the CDPK domains within the sequences. The aa count, molecular weight (kDa), and other properties of the CDPK proteins were assessed utilizing the “ProtParam tool” (https://web.expasy.org/protparam/). The phylogenetic studies employed the Neighbor-Joining (NJ) technique with a bootstrap value of 1,000 replicates. The ClustalW algorithm was employed to align the PvCDPK protein sequences (Thompson et al., 1997). Evolutionary diagrams were produced via MEGA v7 (Tamura et al., 2011). The iTOL database was utilized to construct the phylogenetic tree (Letunic & Bork, 2011).

Structure, chromosomal localization, duplication and comparative analysis of PvCDPK gene family members

The coding and non-coding sections of the PvCDPK genes were retrieved utilizing the Gene Structure Display v2.0 web tool using the genomic and coding sequence (CDS) sequences (http://gsds.gao-lab.org/) (Hu et al., 2015). The positions of PvCDPK genes on the chromosome were derived from the Phytozome v12.1 database (https://phytozome-next.jgi.doe.gov/). PvCDPK genes were delineated on each chromosome of P. vulgaris L. utilizing MapChart (Voorrips, 2002). MCScanX (The Multiple Collinearity Scan Toolkit) (Wang et al., 2012) was used with default settings to determine the orthologous relationship between CDPK genes of P. vulgaris L. and G. max.

The substitution ratios (Ka, Ks, and Ka/Ks) between duplicate pairs of PvCDPK genes were estimated using PAL2NAL (http://www.bork.embl.de/pal2nal/#Ref) (Suyama, Torrents & Bork, 2006) and the AML interface tool (https://github.com/abacus-gene/paml) (Yang, 2007). To estimate the duplication and divergence time (MYA) of CDPK genes, synteny maps were generated using TBtools (Chen et al., 2020), and the formula T = Ks/2λ (λ = 6.56E−9) was applied (Yang & Nielsen, 2000; Lynch & Conery, 2003).

To uncover more conserved PvCDPK protein motifs, the Multiple Em for Motif Elicitation (MEME Tool) (https://nbcr.net/meme/meme/meme-download.html) was used (Bailey et al., 2006). The parameters 2, 50, and 10 were used for minimum and maximum width and maximum number of motifs, respectively. The motif regions were set in the range of 2–300. Area distribution repetitions might be any number. Motifs were analyzed with the InterPro database as outlined by Quevillon et al. (2005). The WEBLOGO online web tool (http://weblogo.berkeley.edu/logo.cgi) produced CDPK domain sequence logos for conserved area sequence analysis (Crooks et al., 2004).

Subcellular localization and analysis of cis-acting elements of PvCDPK gene family

The upstream sections (Table S1) containing two kilobase pair (kbp) DNA segments of each PvCDPK gene family member were analyzed using the PlantCARE (http://bioinformatics.psb.ugent.be/webtools/plantcare/html/) database for cis-acting element analysis (Lescot, 2002). A figure showing cis-acting elements was generated using TBTools (Chen et al., 2020). WoLF PSORT (https://wolfpsort.hgc.jp) predictor was used to predict the subcellular localization of PvCDPK proteins (Horton et al., 2007).

Common bean homology modeling for CDPK proteins

The Phyre2 database (http://www.sbg.bio.ic.ac.uk/ phyre2/html/page.cgi?id=index) was used to acquire the 3D structures, and protein homology modeling was obtained using previously identified CDPK protein sequences (Kelley et al., 2015). The best 3D image was obtained by comparing the protein models’ reliability rates.

An analysis of the ontology of genes and the links between CDPK proteins in P. vulgaris L.

Protein–protein interactions were examined to ascertain their functional and physical relationships using the STRING database (https://string-db.org). The obtained information was categorized and integrated with the confidence level for every interaction between proteins. Cytoscape software was used to visualize the interactions of CDPK proteins, both among themselves and with other proteins (Shannon et al., 2003). The implementation of functional genomics techniques is a crucial prerequisite for the functional annotation of novel sequence data in plant biotechnology research. Ontology data for PvCDPK genes were acquired using the Blast2GO program, and this information was utilized to assess the functional characteristics of PvCDPK proteins (Conesa et al., 2005).

In silico gene expression analysis

RNA-seq data of P. vulgaris L. under salt and drought stress was taken from the National Center for Biotechnology Information (NCBI)’s SRA collection. The used accession numbers were SRR957668 (leaf subjected to salt stress), SRR958469 (leaf salt control) (Hiz et al., 2014), SRR8284481 (leaf subjected to drought stress), and SRR8284480 (leaf drought control). Gene expression data were normalized utilizing reading per kilobase of transcript per million mapped reads (RPKM) (Mortazavi et al., 2008). The Orange software (Demsar et al., 2013) was employed to transform the RPKM data to log2 and generate a heatmap.

Experimental plant materials and treatments

The P. vulgaris L. cultivars “Elkoca-05” and “Serra” employed in this study were obtained from the Molecular Biology and Genetics Department, Erzurum Technical University. The genotype-specific seeds underwent surface sterilization for 5–7 m with a 1% (v/v) NaOCl solution. Subsequently, perlite was utilized for the germination process. The seedlings were relocated to a hydroponic medium comprising 0.2 L of modified 1/10 Hoagland solution upon attaining the developmental stage specified by Büyük et al. (2019). P. vulgaris L. seedlings were grown at 25 °C and 70% relative humidity in a controlled cultivation room with light and a photosynthetic photon flux of 250 mmol m−2 s−1. After common bean seedlings reached the first trifoliate stage in the growth chamber, the control group was treated with 0 mM NaCl, and the stress treatment group was subjected to salt stress for nine days using Hoagland’s solution and 150 mM NaCl (for medium salinity stress). Concurrently, drought-stressed common bean plants grown in the same conditions were kept for 24 h in a Hoagland solution that was treated with either 0 (control) or 20% PEG6000 (Aygören et al., 2023). Root and leaf tissues from two distinct common bean cultivars were collected following the ninth day of stress treatment. Following the specified duration, the leaf tissue of the common bean genotypes was stored in liquid nitrogen and preserved at −80 °C until the analysis was performed. Three biological replicates of the common bean genotypes utilized in the study were cultivated, and these replicates were employed for qRT-PCR analysis. The root and leaf tissues were subjected to distinct qPCR analyses.

In vitro qRT-PCR analysis

Trizol Reagent (Invitrogen Life Technologies, Waltham, MA, USA) was used to extract total RNAs. The Multiskan Go spectrophotometer (Thermo Fisher Scientific, Waltham, MA, USA) was employed to quantify RNA, while a 1.5% agarose gel was utilized to evaluate the quality of the sample. To perform complementary DNA synthesis, the SensiFAST cDNA Synthesis Kit (Cat No: Bio-65053, UK) was utilized, following the instructions provided by the manufacturer. The qRT-PCR study was focused on five PvCDPK genes that were selected from the RNA-seq data. The qRT-PCR reactions were conducted using the RotorGene Q Real-Time PCR System (Corbett Research, Qiagen GmbH, Hilden, Germany) and ABT SYBR Green Mix (Cat. No.: Q03-02-01, Ankara, Turkey). A total of 20 µL of qRT-PCR mix was used, including 10 µL of ABT SYBR Green Mix (2x), 0.4 µL of each primer (one µM forward and reverse), and 200 ng of cDNA. The reaction was carried out as follows: 10 min at 95 °C to be 1 cycle; 15 s at 94 °C, 30 s at 60 °C, and 30 s at 72 °C to be 40 cycles.

Statistical analyses

The housekeeping gene used was the β-actin gene from P. vulgaris L. The 2−ΔΔCT technique for relative quantification was used to standardize the qRT-PCR data (Livak & Schmittgen, 2001). Information on the primers used in this study is presented in Table S2. A two-way analysis of variance (ANOVA) with Dunnett’s test at the 0.05 significant level was utilized to conduct statistical studies in GraphPad Prism 7.

Results

Identification and physicochemical characteristics of CDPK gene family in P. vulgaris L.

Here, 25 CDPK s were identified in the P. vulgaris L. genome through bioinformatics tools and renamed PvCDPK1.1 to PvCDPK32 according to their locations on chromosomes. The amino acid number, protein molecular weight, and theoretical pI (isoelectric point) of the proteins were identified (Table 1). The number of amino acid sequence lengths of PvCDPK s ranged from 298 (PvCDPK4.2) to 582 (PvCDPK2). The molecular weights ranged from 33.43 kDa (PvCDPK4.2) to 65.13 kDa (PvCDPK2), and the pI values ranged from 4.82 (PvCDPK4.2) to 9.21 (PvCDPK16).

Table 1 Characteristics of CDPK proteins in the P. vulgaris genome.

PvGene name	Amino acid	MW (kDa)	pI	EF-hand no	N- myrist	N- palmit	Localization	
PvCDPK1.1	575	64.55	5.09	3	No	Yes	chlo	
PvCDPK1.2	581	64.99	5.28	4	No	Yes	chlo	
PvCDPK2	582	65.13	5.8	4	No	Yes	chlo	
PvCDPK3.1	502	56.71	6.05	4	Yes	Yes	chlo	
PvCDPK3.2	519	58.28	5.9	4	Yes	Yes	mito	
PvCDPK4.1	491	55.16	5.43	4	No	Yes	mito	
PvCDPK4.2	298	33.43	4.82	4	No	Yes	cyto	
PvCDPK6	562	63.01	5.57	4	No	Yes	nucl	
PvCDPK8	516	58.68	6.5	2	No	Yes	chlo, mito	
PvCDPK9	525	59.10	6.3	4	Yes	Yes	cyto	
PvCDPK10.1	537	60.89	5.96	4	No	Yes	cyto	
PvCDPK10.2	550	61.99	6.09	4	No	Yes	cyto	
PvCDPK11.1	505	56.93	5.24	4	No	Yes	chlo	
PvCDPK11.2	496	55.76	5.32	4	No	Yes	pero	
PvCDPK13	531	59.76	5.86	3	No	Yes	cyto	
PvCDPK16	569	64.69	9.21	4	Yes	Yes	chlo	
PvCDPK17.1	544	60.54	5.09	4	Yes	Yes	cyto	
PvCDPK17.2	521	58.49	5.58	4	Yes	Yes	cyto	
PvCDPK20	582	64.79	5.35	4	No	Yes	chlo	
PvCDPK21	546	60.72	5.76	4	Yes	Yes	cyto	
PvCDPK24	539	61.07	6.46	4	Yes	Yes	chlo	
PvCDPK28	527	59.72	8.92	4	No	Yes	chlo	
PvCDPK29.1	527	60.18	6.11	4	Yes	Yes	cyto_nucl	
PvCDPK29.2	511	57.77	5.64	2	No	Yes	cyto	
PvCDPK32	538	60.98	6.29	3	No	Yes	cyto	
Notes.

pI Theoretical isoelectric point

EF-hand EF-hand calcium-binding domain

N-myrist myristoylation

N-palmit palmitoylation

chlo chloroplast

mito mitochondrion

cyto cytosol

nucl nuclear

pero peroxysome

Chromosomal location and duplication events

The 25 PvCDPK genes were distributed unevenly on eight chromosomes: Chr1 (four genes), Chr2 (four genes), Chr3 (three genes), Chr6 (one gene), Chr7 (six genes), Chr8 (four genes), Chr9 (two genes), and Chr11 (one gene) but not on the other chromosomes (4, 5, 6, and 10) of the common bean (Table 2).

Table 2 CDPK gene family members of P. vulgaris L. with Arabidopsis orthologs, chromosome locations, gene start and end point.

PvGene name	Arabidopsis ortholog locus	Phytozome ID	Chr name	Strand	Gene start (bp)	Gene end (bp)	
PvCDPK1.1	AT5G04870 (AtCPK1)	Phvul.001G197700.1	PvChr1	Forward	45,693,570	45,697,541	
PvCDPK1.2	AT5G04870 (AtCPK1)	Phvul.007G233900.1	PvChr7	Forward	35,767,288	35,774,697	
PvCDPK2	AT3G10660 (AtCPK2)	Phvul.007G266100.1	PvChr7	Reverse	38,724,008	38,729,058	
PvCDPK3.1	AT4G23650 (AtCPK3)	Phvul.002G161200.1	PvChr2	Reverse	31,597,834	31,604,224	
PvCDPK3.2	AT4G23650 (AtCPK3)	Phvul.002G294500.1	PvChr2	Reverse	46,343,902	46,352,694	
PvCDPK4.1	AT4G09570 (AtCPK4)	Phvul.007G089200.1	PvChr7	Forward	9,217,140	9,221,899	
PvCDPK4.2	AT4G09570 (AtCPK4)	Phvul.007G089301.1	PvChr7	Forward	9,223,593	9,231,935	
PvCDPK6	AT2G17290 (AtCPK6)	Phvul.008G292500.1	PvChr8	Forward	62,964,859	62,973,752	
PvCDPK8	AT5G19450 (AtCPK8)	Phvul.007G253300.1	PvChr7	Reverse	37,518,913	37,524,077	
PvCDPK9	AT3G20410 (AtCPK9)	Phvul.008G266600.1	PvChr8	Reverse	61,203,175	61,207,570	
PvCDPK10.1	AT1G18890 (AtCPK10)	Phvul.003G194100.1	PvChr3	Reverse	41,784,590	41,788,241	
PvCDPK10.2	AT1G18890 (AtCPK10)	Phvul.009G190566.1	PvChr9	Reverse	28,940,689	28,945,605	
PvCDPK11.1	AT1G35670 (AtCPK11)	Phvul.002G279300.1	PvChr2	Forward	44,866,006	44,871,378	
PvCDPK11.2	AT1G35670 (AtCPK11)	Phvul.009G160100.1	PvChr9	Reverse	23,666,550	23,672,870	
PvCDPK13	AT3G51850 (AtCPK13)	Phvul.008G098400.1	PvChr8	Forward	10,295,962	10,304,546	
PvCDPK16	AT2G17890 (AtCPK16)	Phvul.002G108700.1	PvChr2	Reverse	23,210,281	23,216,106	
PvCDPK17.1	AT5G12180 (AtCPK17)	Phvul.006G015300.1	PvChr6	Forward	7,024,909	7,028,768	
PvCDPK17.2	AT5G12180 (AtCPK17)	Phvul.008G201900.1	PvChr8	Forward	54,865,627	54,868,891	
PvCDPK20	AT2G38910 (AtCPK20)	Phvul.007G265100.1	PvChr7	Reverse	38,623,002	38,629,456	
PvCDPK21	AT4G04720 (AtCPK21)	Phvul.003G078400.1	PvChr3	Forward	12,570,425	12,575,047	
PvCDPK24	AT2G31500 (AtCPK24)	Phvul.011G055400.1	PvChr11	Forward	4,877,733	4,881,802	
PvCDPK28	AT5G66210 (AtCPK28)	Phvul.003G261700.1	PvChr3	Reverse	50,115,856	50,126,462	
PvCDPK29.1	AT1G76040 (AtCPK29)	Phvul.001G002800.2	PvChr1	Forward	165,997	170,716	
PvCDPK29.2	AT1G76040 (AtCPK29)	Phvul.001G002900.1	PvChr1	Forward	171,533	174,640	
PvCDPK32	AT3G57530 (AtCPK32)	Phvul.001G135300.1	PvChr1	Reverse	37,388,614	37,393,061	

The investigation of gene duplication revealed that the PvCDPK1.1/PvCDPK2 and PvCDPK3.1/PvCDPK3.2 genes had a synonymous substitution (Ks) value of 0.64, the PvCDPK8/PvCDPK32 genes had a Ks value of 0.81, and the PvCDPK10.1/PvCDPK10.2 genes with 0.86 Ks, PvCDPK11.1/PvCDPK11.2 genes with 0.74 Ks, and PvCDPK17.1/ PvCDPK17.2 genes with 0.69 Ks. PvCDPK4.1/PvCDPK4.2 and PvCDPK29.1/PvCDPK29. 2 genes were found to be tandemly duplicated with Ks values of 0.42 and 1.11, respectively (Table 3). These genes had Ka/Ks ratios ranging from 0.08 to 0.28. Natural selection during duplication events is represented by values equal to 1, purifying selection is indicated by values less than 1, and positive selection in the evolutionary process is shown by Ka/Ks values greater than 1 (Juretic et al., 2005). Accordingly, it can be inferred that all PvCDPK genes have undergone purifying selection (Fig. 1). In addition, the differentiation time of six pairs of segmentally duplicated genes ranged from 48.94 million years ago (MYA) to 65.57, while tandem pairs ranged from 32.09 to 84.95 MYA.

Table 3 Duplication events and evolutionary analysis of PvCDPK genes.

Gene1	Gene2	Ka	Ks	Ka/Ks	MYA	Duplication type	
PvCDPK1.1	PvCDPK2	0.17	0.64	0.27	49.11	Segmental	
PvCDPK3.1	PvCDPK3.2	0.09	0.64	0.14	48.94	Segmental	
PvCDPK8	PvCDPK32	0.10	0.81	0.12	61.57	Segmental	
PvCDPK10.1	PvCDPK10.2	0.07	0.86	0.08	65.30	Segmental	
PvCDPK11.1	PvCDPK11.2	0.08	0.74	0.11	56.19	Segmental	
PvCDPK17.1	PvCDPK17.2	0.09	0.69	0.13	52.52	Segmental	
PvCDPK4.1	PvCDPK4.2	0.12	0.42	0.28	32.09	Tandem	
PvCDPK29.1	PvCDPK29.2	0.25	1.11	0.22	84.95	Tandem	
Notes.

MYA million years ago

Figure 1 Distribution of PvCDPK genes on P. vulgaris chromosomes.

Black lines represent segmental duplicated genes and red lines represent tandem duplicated genes.

Phylogenetic relationships and synteny analysis of PvCDPKs in P. vulgaris L. and different plants

Using the protein sequences from the common bean, A. thaliana, and G. max, a phylogenetic tree was created to ascertain the phylogenetic relationships for the common bean’s CDPK gene family. In the phylogenetic analysis, researchers used a total of 98 protein sequences: 25 from P. vulgaris L., 34 from Arabidopsis, and 39 from G. max. As illustrated in Fig. 1, the genes were clustered into three major subfamilies: A, B, and C. Group A is the largest with 69 genes, while Group C is the smallest with nine genes. PvCDPK4.2 was noticed relatively independently of other PvCDPKs. Within the three groups formed in this phylogenetic tree, which played a crucial role in elucidating the molecular evolutionary process, PvCDPK genes were observed to be homologously distributed, especially with AtCDPK genes (Fig. 2).

Figure 2 Phylogenetic analysis of CDPK proteins from A. thaliana (34), G. max (39) and P. vulgaris (25).

A. thaliana is represented by a green circle, G. max by a purple square and P. vulgaris by a maroon star. G. max sequences were obtained from Liu et al. (2016) and used. The locations of 69 genes in group (A), 20 genes in group (B) and nine genes in group (C) are shown.

Synteny analysis was performed to examine shared structural changes in the genome, including chromosomal fission and fusion. The analyses revealed 57 syntenic relationships between P. vulgaris L. and G. max and 23 syntenic relationships between P. vulgaris L. and A. thaliana. While a syntenic relationship was found between all PvCDPK genes and G. max genes, no syntenic relationship was found between AtCDPK genes and PvCDPK genes only in PvChr-6. This indicates a strong evolutionary similarity between G. max and P. vulgaris L. It can also be said that there is a strong syntenic relationship between A. thaliana and P. vulgaris L. in terms of chromosomal significance. In addition, CDPK genes are equally distributed in these genomes, indicating that these gene pairs are widely distributed within the genomes (Fig. 3).

Figure 3 Synteny analysis between A. thaliana, G. max and P. vulgaris CDPK genes.

The red lines highlight the syntenic gene pairs between bean and Arabidopsis, while blue lines highlight the syntenic gene pairs with G. max.

Gene structure and motif analysis of PvCDPK gene family

The analysis of gene structure revealed that the 25 PvCDPK gene family members together harbor 169 introns and 195 exons. PvCDPK16 and PvCDPK28 have 11 introns; PvCDPK3.1, PvCDPK3.2, PvCDPK8, PvCDPK9, PvCDPK17.1, PvCDPK17.2, PvCDPK21, PvCDPK24, PvCDPK29.1, PvCDPK29.2, and PvCDPK32 have seven introns; PvCDPK1.1, PvCDPK1.2, PvCDPK2, PvCDPK4.1, PvCDPK6, PvCDPK10.1, PvCDPK10.2, PvCDPK11.1, PvCDPK11.2, PvCDPK13, and PvCDPK20 have six introns; and PvCDPK4.2 has five introns (Fig. 4).

Figure 4 Structural representation of PvCDPK genes.

Maroon color represents exon and black lines represent intron regions. Sand colored parts represent 5′ and 3′ UTR regions. The scale bar indicates 10 kb.

A total of 10 conserved motifs were identified with lengths ranging in length from eight to 50 amino acids using MEME (Fig. 5; Table S1). All proteins were found to contain motifs 2, 5, and 7, while additional motifs were detected solely in particular subgroups (Fig. 4). As an example, motif 8 is only found in one subgroup (Fig. 4). Except for PvCDPK16 and PvCDPK28, all PvCDPKs have four motif 2.

Figure 5 Conserved motif analysis in PvCDPK proteins.

Analysis of PvCDPKs promoter cis-elements

The regulatory mechanisms of PvCDPK genes were explored by analyzing the two kb upstream sequences from their start codons for cis-regulatory element composition.

As a direct result, a total of 341 cis-acting regulatory elements were identified in the promoters of PvCDPK genes (Fig. 6; Table S1). The promoters of PvCDPK genes also contained a total of 15 cis- acting regulatory elements. The cis-acting elements were divided into four main categories: abiotic/biotic stress-responsive, including 12 elements (ABRE, MYB, MBS, LTR, etc.); development, including elements (CCAAT-box), core elements, and binding sites (W box); and hormonal-responsive (as-1), including only one element.

Figure 6 Cis-acting element analysis of PvCDPK genes.

The elements on the genes represent those that play a role during plant stress. The different colors of the lines indicate the various cis-acting elements within the two kb promoter region located upstream of the PvCDPK gene.

The greatest number of cis-acting elements were determined in MYC and MYB, and these elements, together with the MBS element, are associated with drought stress defense. While MYC and MYB were found in all genes, MBS was found to be associated with PvCDPK-1.2, -3.1, -4.1, -4.2, -9, -10, -11.1, -13, -17.1, -20, -29.1, and -29.2 genes. Biotic and abiotic stress is represented by the highest number of elements, indicating PvCDPK genes have important roles in response to biotic and abiotic stress (Fig. 6; Table S1). Considering this viewpoint, it is likely that these PvCDPKs participate in multiple biological functions.

Homology modeling, 3D structural prediction, protein interaction network construction, and gene ontology annotation of PvCDPK proteins

Proteins interact to fulfill their functions; therefore, comprehending the connections and mechanisms of complex biological processes is essential. PvCDPK protein sequences were utilized to ascertain their protein-protein interactions via the STRING interface. Here, it was found that 25 proteins interacted with five different common bean proteins. These proteins were V7BRL4_PHAVU-Phvul.006G142500 and V7BS13_PHAVU-Phvul.006G157600 (calcium binding protein 39), V7BH71_PHAVU-Phvul.007G22340 (heat stress transcription factor A-3), V7CME5_PHAVU-Phvul.002G160700 (respiratory burst oxidase homolog protein F-related), and V7CPP5_PHAVU-Phvul.002G293700 (PTHR11972//PTHR11972:SF81 - NADPH oxidase) (Fig. 7; Table S1). All PvCDPK proteins interacted with V7CME5_PHAVU-Phvul.002G160700. PvCDPK17.1 and PvCDPK17.2 interacted with PvCDPK24. The remaining proteins showed no interaction with each other.

Figure 7 Interaction analysis of PvCDPK proteins both among themselves and with other proteins.

Gene ontology facilitates the comprehension of genes through the analysis of biological processes, molecular functions, and cellular components. In the biological process, PvCDPK genes are enriched in the peptidyl-serine phosphorylation, protein autophosphorylation, and intracellular signal transduction. The cellular component category included the nucleus, cytoplasm, and intracellular anatomical structure. Protein serine kinase activity, calcium ion binding activity, calmodulin-dependent protein kinase activity, calmodulin binding activity, and calcium-dependent protein serine/threonine kinase activity were categorized in molecular function (Fig. 8; Table S1). CDPK proteins were identified using the Phyre2 database, and homology modeling was visualized via the 3D modeling technique. Figure 9 illustrates the three-dimensional homology models of the proteins identified in the study.

Figure 8 Gene ontology analysis of PvCDPK proteins.

The cellular component in which it is found, the biological process in which it is involved and the molecular functions it shows are included.

Figure 9 3D homology models of PvCDPK proteins using Phyre2 database and by 3D modelling.

Models were visualized using rainbow colors from N to C terminus.

In silico expression profiles of PvCDPK gene family drought and salt stress

Throughout their period of development and growth, plants are greatly impacted by a wide variety of environmental conditions, including low temperatures, high salt, and drought. Expression pattern analysis can help understand the biological functions of PvCDPK in tissue-specific or abiotic stresses such as salt and drought. To comprehensively analyze the mRNA expressions of PvCDPK genes, RNA-seq data from four normal and treatment samples from the NCBI SRA database were obtained, and FPKM values of 25 PvCDPK genes were evaluated. Five different tissues were taken for evaluation in this study. All the PvCDPK genes were expressed in at least one tissue. Different PvCDPK genes revealed differential expression patterns. For example, only PvCDPK17.1 and PvCDPK24 displayed expression in flowers but no other tissues. PvCDPK11.2 was the most expressed PvCDPK in leaves and stem, whereas PvCDPK6 was significantly expressed in flowers, nodules, root, and stem (Fig. 10A; Table S1). With these findings, it can be said that CDPK genes actively contribute to common bean organ development.

Figure 10 In silico analysis of PvCDPK gene expression in (A) different tissues (root, stem, nodule, leaf, and flower), and (B) under drought and salt stress conditions compared to controls.

In this study, in silico gene expression analysis under drought and salt stresses was determined. PvCDPK16 and PvCDPK6 expressions were higher in control plants compared to drought- and salt-treated plants. PvCDPK11.2, PvCDPK10.2, PvCDPK32, PvCDPK21, PvCDPK13, and PvCDPK10.1 genes expressed higher than control under drought stress. However, these gene expressions showed lower expression under salt stress (Fig. 10B; Table S1). PvCDPK28 and PvCDPK3.2 were induced after salt treatments, but their expressions reduced after drought treatment. No important change was determined in the expression patterns of other genes. On the other hand, among these genes, only PvCDPK10.2 expression displayed the same expression level between control and treated plants. These findings suggest that PvCDPKs may be involved in the response to a range of abiotic stresses, with different genes displaying distinct responses to stress.

qRT-PCR analyses

In this study, common bean seedlings were treated with drought and salt to examine the function of PvCDPK gene members. Here, two cultivars, Elkoca-05 and Serra, were used. qRT-PCR analyses were performed for five specific primers (PvCDPK1, PvCDPK4, PvCDPK10, PvCDPK20, and PvCDPK29) designed using RNA-seq data, and the results are shown in Fig. 11. Firstly, it was determined that no non-specific results were obtained in the negative control analyses performed in qPCR. Under drought stress, the PvCDPK1, PvCDPK4, PvCDPK10, and PvCDPK29 gene expressions were increased in Elkoca-05. However, there was no significant change in the expressions of PvCDPK analyzed, nor of the PvCDPK genes in Serra, neither in leaf nor in root.

Figure 11 Expression analysis of PvCDPK genes under salt an drought stresses in two cultivars’ root and leaf (Serra and Elkoca-05) using qRT–PCR method.

*p < 0.05, **p < 0.01, ***p < 0.001, ****p < 0.0001, ns, non-significant.

Under salt stress, different expression patterns were observed compared to drought stress. PvCDPK1 and PvCDPK29 were induced in leaf, while PvCDPK4 was induced in root in the Serra cultivar. Besides, PvCDPK4 expression also increased in Elkoca-05 root. No significant change was observed for PvCDPK4 expression in leaf under salt stress in both Serra and Elkoca-05.

Consequently, although gene expression levels of the PvCDPK gene family varied among cultivars, the same genes analyzed under drought stress exhibited no variation in the Elkoca-05 cultivar across different tissues. Although the expression levels of PvCDPK1 and PvCDPK29 in the Elkoca-05 cultivar increased in drought stress treatment, their expression decreased in salt treatment in all tissues. PvCDPK4 was induced under both drought and salt stress in both two cultivars. These findings are in agreement with in silico analyses.

Discussion

Phaseolus vulgaris L. (common bean), a legume species of high economic importance, serves as a major dietary protein and nutrient source globally. However, abiotic stressors, particularly salinity and drought, pose significant threats to its yield and productivity. While previous genome-wide studies in P. vulgaris L. have identified various gene families (Büyük et al., 2019; Akbulut et al., 2022; Aygören et al., 2022; De Souza Resende, Santos & De Souza, 2022; Muslu et al., 2023; Aygören et al., 2023; Chakraborty et al., 2023; Kasapoğlu et al., 2024), comprehensive investigations on Ca2+-related gene families remain limited. This study presents the inaugural genome-wide identification and systematic characterization of the CDPK gene family in P. vulgaris L. The results offer significant insights into the possible roles of the gene family in evolutionary conservation and stress responses. CDPKs, similar to transcription factors, are key regulators of gene expression and mediate diverse physiological responses through calcium signaling. These kinases are activated by intracellular Ca2+ fluctuations and play essential roles in the perception and transduction of abiotic stress signals. Previous reports have documented varying numbers of CDPK genes across plant species, ranging from 11 to 85 (Cheng et al., 2002; Ray et al., 2007; Wang et al., 2016; Crizel et al., 2020; Fan et al., 2023; Linghu et al., 2023; Yang et al., 2023; Burra, Rajeshwari & Suma, 2023; Xiong et al., 2024). In this study, 25 PvCDPK genes were identified, a number consistent with findings in related species.

Phylogenetic analysis revealed that the PvCDPK proteins clustered into three main clades together with their A. thaliana homologs, indicating a high degree of evolutionary conservation. Nonetheless, the autonomous clustering of certain members, such as PvCDPK4.2, indicates that these genes may have developed species-specific functions. This facilitates the formulation of new hypotheses suggesting that PvCDPK4.2 may play a role in particular developmental processes (e.g., flowering or symbiotic nitrogen fixation) in P. vulgaris L., apart from abiotic stress response. Moreover, structural analyses showed that the PvCDPK genes collectively contained 169 introns and 195 exons. The diversity in exon-intron architecture suggests functional divergence and evolutionary adaptation within the gene family.

Subcellular localization predictions indicated that most PvCDPK proteins are cytosolic, although several members are also localized to the nucleus, chloroplast, peroxisome, and mitochondria. These findings are partially consistent with previous reports in Fragaria× ananassa (strawberry), where CDPKs localize to the plasma membrane, cytoplasm, nucleus, and chloroplast (Crizel et al., 2020). In contrast, CDPKs in Gossypium hirsutum (cotton) were predominantly localized in the nucleus (Lv et al., 2024), highlighting species-specific differences in subcellular distribution and potentially distinct physiological roles.

Genomic mapping revealed a wide distribution of 25 PvCDPK genes across eight chromosomes. Notably, chromosomes Chr1, Chr2, Chr7, and Chr8 harbored a higher density of CDPK loci, whereas others such as Chr4, Chr5, Chr6, and Chr10 lacked any PvCDPK genes. Similarly, 84 CDPK genes were identified as being distributed on 26 chromosomes in G. barbadense, whereas in tomato, 29 CDPK genes were found to be distributed on 12 chromosomes (Hu et al., 2016; Shi & Zhu, 2022). The genes in the PvCDPK family have undergone expansion via regional and tandem duplications, yet all have persisted under purifying selection (Ka/Ks < 1); consequently, these genes are functionally significant and conserved. These distribution patterns likely indicate chromosomal rearrangements and duplications that facilitated the diversification of this gene family throughout evolution (Cortés et al., 2018).

Promoter analysis revealed that PvCDPK genes contain a large number of stress-related regulatory elements such as MYB, MBS, ABRE, and LTR. The presence of a large number of these elements suggests that CDPK genes have the potential to respond not only to salt and drought but also to other environmental influences such as cold and oxidative stress (López-Hernández et al., 2023; Han et al., 2024). MYC and MYB motifs were detected in the promoter regions of all PvCDPK genes, while the MBS element was found in 12 genes. These findings are consistent with prior studies in Ipomoea batatas (sweet potato) and wheat, where these elements have been linked to abiotic stress tolerance (Li et al., 2022a; Li et al., 2022b; Linghu et al., 2023), suggesting that PvCDPKs may act as upstream regulators in stress-responsive transcriptional networks.

Gene ontology (GO) enrichment analysis gave new information about the functional roles of PvCDPK genes. Biological process annotations included peptidyl-serine phosphorylation, autophosphorylation, and intracellular signal transduction. Molecular function categories were dominated by protein serine/threonine kinase activity, calcium ion binding, and calmodulin-dependent protein kinase activity. Cellular component classifications indicated nuclear, cytoplasmic, and organelle-associated localizations. These functional predictions are consistent with recent findings in other plant systems (Li et al., 2022a; Li et al., 2022b).

Expression profiling using RNA-seq datasets demonstrated that PvCDPK genes exhibit distinct tissue-specific expression patterns. For instance, PvCDPK17.1 and PvCDPK24 were exclusively expressed in floral tissues, whereas PvCDPK11.2 showed high expression in leaves and stems. PvCDPK6 displayed elevated expression in flowers, nodules, roots, and stems. These results support the hypothesis that CDPKs function in organ development, tissue differentiation, and stress adaptation. This is in agreement with previous reports implicating CDPKs in root development, pollen maturation, and phytohormone signaling pathways (Li et al., 2018; Wen et al., 2020; Li et al., 2022a; Li et al., 2022b). Particularly, the high expression of PvCDPK6 in flower, tuber, root, and shoot raises the hypothesis that this gene may be associated with developmental transitions and hormone signaling. This study suggests that PvCDPK6 may be a regulator that responds to growth regulators such as jasmonate or gibberellin (Xu & Huang, 2017).

To further validate these findings, qRT-PCR analyses were conducted to assess PvCDPK gene expression under drought and salinity stress conditions in two P. vulgaris L. cultivars. The results revealed cultivar-specific and stress-dependent expression dynamics. In particular, PvCDPK1 and PvCDPK29 were significantly upregulated in Elkoca-05 under drought stress but were downregulated in all tissues under salinity stress. PvCDPK1 and PvCDPK29 were exclusively up-regulated in Elkoca-05 in response to drought, suggesting that these genes are regulated by distinct mechanisms influenced by genetic background. These differences are believed to represent the molecular foundation for the variation in abiotic stress tolerance among cultivars. This finding supports the idea that PvCDPK genes can be used as target gene candidates in the development of lines with high stress tolerance through biotechnological applications. Notably, PvCDPK4 was induced under both drought and salt treatments in both cultivars. In support of this, studies conducted in the Colombia region performed comparative genome-wide association studies (GWAS) between drought- and heat-susceptible common beans and drought-tolerant tepary beans, identifying several genes directly associated with drought tolerance responses, as well as genes related to morphological, physiological, and metabolic adjustments, signal transduction, and fatty acid and phospholipid metabolism (Burbano-Erazo et al., 2021; López-Hernández et al., 2023). Incorporating stress-tolerant cultivars into future analyses, including genome-wide association studies, is essential for improving the comparability of gene expression data under abiotic stress conditions. Plants under drought stress close their stomata to minimize water loss (Tahir, Karagiannis & Tian, 2022). CDPK genes contribute to stress tolerance by being involved in ABA signaling pathways and ROS scavenging (Asano et al., 2012). These genes were found to regulate stomatal closure through ABA (Mori et al., 2006). ABA treatment was also found to increase the expression of Setaria italica CDPK genes (Yu et al., 2018). Based on this information, the interaction between ABA-CDPK genes in drought stress is thought to be important in stress tolerance (Shu et al., 2020). The increased expression of PvCDPK genes after drought treatment also indicates that this interaction occurs. The presence of abscisic acid responsive element (ABRE) in the promoter region of the PvCDPK1, PvCDPK4, and PvCDPK10 genes that respond to drought stress also makes sense of this relationship. As a result of salt stress, Na+ and K+ balance is disrupted. In order to restore this balance, S-ABA treatment was applied to O. sativa under salt stress, and as a result of the treatment, the K+ content of the plant increased and the ion homeostasis came into balance, resulting in increased antioxidant enzyme activity (Jiang et al., 2024). In salt-treated Lolium perenne, it was found that Na+/K+ homeostasis was achieved by observing increased K+ content in LpCDPK27 overexpressed lines (Chen et al., 2025). These results suggest that increased expression of PvCDPK genes in salt stress is a response to stress. These results corroborate the RNA-seq data and emphasize the regulatory role of PvCDPKs in abiotic stress responses.

Collectively, this study presents the first comprehensive characterization of the CDPK gene family in P. vulgaris L., offering novel insights into their structure, evolution, regulatory potential, and functional relevance under stress conditions. The results establish a significant basis for forthcoming functional genomic and molecular breeding initiatives focused on improving stress tolerance in common beans. However, although the expression analysis of a limited number of genes performed using qRT-PCR in this study supports the results of the bioinformatic analysis, it is not possible to generalize to all members of the gene family. Moreover, it should not be overlooked that the potential limitations arising from seed uniformity must also be considered (Peláez et al., 2022). To better understand the roles of these genes in different tissues, developmental stages, and other abiotic stress conditions, further functional validation studies, such as expression profiling and knockout and overexpression, are needed.

Conclusion

Using current databases and tools, CDPK genes belonging to the P. vulgaris L. species were identified and characterized across the genome. A total of 25 CDPK genes were identified, distributed across eight chromosomes. In silico analyses revealed that gene expression levels varied across different plant tissues; these findings were supported by in vitro expression analyses conducted on two different bean cultivars under salt and drought stress conditions. These investigations uncovered a significant correlation between PvCDPK genes and drought and salinity stress. The function of CDPK genes, essential for vital metabolic processes such as flowering, root development, and fruit maturation in plants, has been examined in common bean plants for the first time.

This comprehensive study on P. vulgaris L. is expected to significantly contribute to breeding research, improve our knowledge regarding stress-related metabolic processes and responses, and serve as a valuable resource for future scientific investigations. Based on this study, PvCDPK4.2 may be involved in species-specific developmental regulation, and silencing this gene with functional genetic approaches such as CRISPR/Cas9 may indicate whether it alters developmental phenotypes. Furthermore, PvCDPK6 may be a node of hormone signaling pathways (e.g., ABA, JA). It should be investigated how the expression of this gene changes with hormone treatments.

Supplemental Information

Supplemental Information 1 Comprehensive Data on CDPK Genes in Phaseolus vulgaris L

Supplemental Information 2 Primer sequences used for qRT-PCR

Supplemental Information 3 Raw data of the qPCR analysis showing the sample list, results, and 2−ΔΔCt values

Supplemental Information 4 MIQE Checklist

Additional Information and Declarations

Competing Interests

Author Contributions

Data Availability

The author declare there are no competing interests.

Gülşen Güçlü conceived and designed the experiments, performed the experiments, analyzed the data, prepared figures and/or tables, authored or reviewed drafts of the article, and approved the final draft.

The following information was supplied regarding data availability:

The raw data is available in the Supplemental Files.

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
