# Peer review of "Genome-wide identification of calcium-dependent protein kinase (CDPK) family members in Phaseolus vulgaris L. and expression analysis during abiotic stresses"

_PeerJ, doi:10.7717/peerj.20217_

## Round 0.1 · original submission · Major Revisions

**Language Note:** The review process has identified that the English language must be improved. PeerJ can provide language editing services - please contact us at [email protected] for pricing (be sure to provide your manuscript number and title). Alternatively, you should make your own arrangements to improve the language quality and provide details in your response letter. – PeerJ Staff

·

Basic reporting

Some changes are necessary to make it easier for the reader.

Experimental design

Feasible.

Validity of the findings

The analyses presented support the study.

Additional comments

Changes and observations are highlighted in the text.

Reviewer 2 ·

Basic reporting

1.1 The authors should be mindful of the tense that is used in the manuscript. They should re-check the whole document for the minor mistakes with regards to the grammar tense, eg, the highlighted sentences in orange refere to change in tense.

Experimental design

On materials and methods, where possible, combine your methods under one sub-heading, to avoid headings with short (one or two sentences) methods.
Line 143: Is the software changing the way the proteins interact with each other? Isn't the stress imposed on the plants the one affecting how the plants react? Can the authors clarify this statement?

Validity of the findings

Line 21-225: The interpretation of the results is all over the place, and confusing the intended message. The authors should rephrase it.
For RNA-seq data, using only one set of results to identify the differential expression is not enough validity, but rather biased. The authors should atleast have used a set with replicates, to validate the expression data.

In the discussion, the authors should try to discuss their results using subtitles, for a good flow of events leading to a conclusion.

Additional comments

The manuscript should be submitted for English editing, for grammar purposes.
More comments can be accessed on the revised copy of the manuscript attached.

Annotated reviews are not available for download in order to protect the identity of reviewers who chose to remain anonymous.

Reviewer 3 ·

Basic reporting

The work “Genome-Wide Identification of Calcium-Dependent Protein Kinase (CDPK) Family Members in Phaseolus vulgaris L. and Expression Analysis During Abiotic Stresses” provides one of the first comprehensive genome-wide identification and characterization of calcium-dependent protein kinase (CDPK) genes in common bean, merging in silico analyses with qRT-PCR validation under drought and salt stress. The identification of 25 PvCDPK genes, their phylogenetic classification, promoter element analysis, and expression profiling provide clues into their potential roles in abiotic stress responses and development. While the work has potential, I do encourage authors to address the following more detailed amendments.

First, the background section in the abstract (L20) and the last paragraph of the introduction (L76) must explicitly incorporate the research gap statement before describing the study goal. Please also make explicit a more concrete scientific hypothesis so that the report reads less descriptive and more hypothesis oriented. Concerning the gap statement, it must reinforce the novelty of the study in the light of existing research in legumes and common bean. Also, please close the latest paragraph in the Introduction by explicitly referring to expected results.

Experimental design

Second, concerning materials candidate orthologous require a more in depth functional annotation using databases like EggNOG, CAZy, KEGG (for instance in L147). Also, please include in L196 a novel subsection entitled “statistical analysis” in which authors must comment of what type of sensitivity, significance, and enrichment tests they performed. How representative is the in silico and experimental sampling is a key question that the power and sensitivity analyses must approach. Please explicitly re-visit these sampling caveats within the discussion section, just before the conclusion paragraph in L436.

Validity of the findings

Third, concerning the results, artwork needs to be better edited. Authors should improve Figures 2, 4 and 6 so that marginal dendrograms show branch support (as a bootstrapping score or alternatively a posterior probability). Also, Figures 9 and 12 are insanely huge with many subpanels across pages, which makes them very hard to read and unprintable. Please amend by reporting more condensed and individualized labeled artwork (i.e., 12 figures in total is too much). Similarly, all figures need high resolution since in the current version they appear pixelated.

Additional comments

Fourth, the discussion, although perceptive, should embrace broader reflections on whether CDPK diversity could be behind the well-known plant yield vs. abiotic tolerance trade-off described in common bean (see Agronomy 2023 13(5):1396 and Agronomy 2021 11:1978 for heat and drought, complementing the discussion in L393), as well as in many other crop species. Besides, since abiotic stresses tolerance could be pleiotropic, please also envision potential concerted responses across the CDPK family. Also related to this point on pleiotropy, authors must also briefly mention that the yield/abiotic tolerance trade-off may also be evident in more subtle ways such as plant architecture and seed nutrient accumulation (see Front Genet 2020 11:656). Any trend among the results pointing towards this direction?

Another key scientific question to be discussed more prominently is how mechanistically interpreting abiotic stress tolerance conferred by the CDPK family could unveil deeper ecological and evolutionary implications (see Front Plant Scie 2018 9:1816). Also concerning the discussion, a major question that authors should prospect in the end of the discussion (for instance complementing L434) is how common bean improvement for abiotic tolerance may unlock and effectively utilize this impressive repertoire of diversity in the CDPK family, ultimately translating this knowledge into the farmer’s field (see Agronomy 2022 12(10):2285). For example, mention more specific pathways that could be targeted for crop improvement. Please expand on these aspects since they would enhance the impact of the study.

A final point concerning the discussion, please describe the major caveats/limitations of the present work (e.g., reliance on in silico data with limited experimental validation of functional predictions) in the closing of the discussion section (L435). It is never beyond the scope of any research to explicitly acknowledge the study caveats. Also link to this point, move the short perspectives section (L436), just before the conclusions in L437. The perspective should also envision what other gene families must future studies like this target in common bean. Also, what about exploring CDPK family diversity across related genepools and species like Tepary bean (Phaseolus acutifolius), which are known to be naturally more tolerant to abiotic stresses? (see Genes 2021 12:556).

---

## Round 0.2 · accepted · Accept

Your revised manuscript is well done. Please make the one change suggested by reviewer 2 (add bootstrap node support in Figure 2), and it will be ready for publication.

Reviewer 3 ·

Basic reporting

I acknowledge the effort made by the authors in revising their manuscript according to my suggestions. I do agree with the carried amendments as well as with the rebuttal. As a last minor detail, please ensure bootstrap node support in Figure 2. This would convey more relevant information concerning the phylogenetic analysis of CDPK proteins in common bean compared to soybean and Arabidopsis thaliana.

Experimental design

-

Validity of the findings

-